

# Impact of grain size and rock composition on simulated rock weathering

Yoni Israeli and Simon Emmanuel

Institute of Earth Sciences, The Hebrew University of Jerusalem, Jerusalem, Israel.

*Correspondence to*: Simon Emmanuel (swemmanuel@gmail.com)

**Abstract.** Both chemical and mechanical processes act together to control the weathering rate of rocks. In rocks with micrometer size grains, enhanced dissolution at grain boundaries has been observed to cause the mechanical detachment of particles. However, it remains unclear how important this effect is in rocks with larger grains, and how the overall weathering rate is influenced by the proportion of high and low reactivity mineral phases. Here, we use a numerical model to assess the effect of grain size on chemical weathering and chemo-mechanical grain detachment. As grain size increases, the weathering rate initially decreases; however, beyond a critical size no significant decrease in the rate is observed. This transition occurs when the density of reactive boundaries is less than ~20% of the entire domain. In addition, we examined the weathering rates of rocks containing different proportions of high and low reactivity minerals. We found that as the proportion of low reactivity minerals increases, the weathering rate decreases non-linearly. These simulations indicate that for all compositions, grain detachment contributes more than 36% to the overall weathering rate, with a maximum of ~50% when high and low reactivity minerals are equally abundant in the rock. This occurs because selective dissolution of the high reactivity minerals creates large clusters of low reactivity minerals which then become detached. Our results demonstrate that the balance between chemical and mechanical processes can create complex and non-linear relationships between the weathering rate and lithology.

## 1 Introduction

The rate at which rocks weather influences the development of landscapes, the formation of soil, and the preservation of buildings and monuments (Brantley, 2008;Putnis and Ruiz-Agudo, 2013;Wilson, 2004). Rock weathering is affected by both chemical and physical processes (Buss et al., 2008;Fletcher et al., 2006;Røyne et al., 2008), that operate from the outcrop scale down to the atomic scale. Although mechanical weathering was typically thought to be important mainly at large scales, enhanced chemical dissolution at grain boundaries allows tiny grains to become mechanically detached from rock surfaces (Silveira and Aarão Reis, 2013;Emmanuel and Levenson, 2014;di Caprio et al., 2016;Krklec et al., 2016;Fischer and Luttge, 2017). This effect was estimated to represent around 40% of the total weathering in fine-grained limestone and is likely to be important in other rock types as well (Levenson and Emmanuel, 2016).

Although grain detachment has been observed in rocks with micrometer size grains, it is unclear how important this effect is in rocks comprising larger grains. From a chemical weathering perspective, a fine-grained rock should dissolve more rapidly than a coarse-grained rock due to its higher reactive surface area (Kirstein et al., 2016;Beckingham et al., 2016).



Moreover, at the nanoscale, tiny grains become intrinsically more reactive than large grains due to thermodynamic and kinetic effects (Emmanuel and Ague, 2011;Briese et al., 2017;Arvidson et al., 2003). Importantly for mechanical detachment, a rock comprising large grains will have a low density of grain boundaries relative to a rock with small grains. Since the grain detachment mechanism involves preferential dissolution along inter-granular boundaries, it might be expected that fine-grained rocks will also be more susceptible to mechanical detachment (Emmanuel and Levenson, 2014). Thus, both the chemical and mechanical effects could make the overall weathering rate strongly dependent on grain size. However, in a study examining experimental weathering rates in carbonate rocks with different textures and crystal sizes, no significant difference was found (Levenson et al., 2015). This apparent lack of dependence on grain size could be due to the inherent difficulty associated with comparing the reaction rates in rocks with varying chemistry and crystal form, and a complete understanding of grain size effects during weathering has yet to be achieved.

Another factor expected to affect weathering rates is the presence of mineral phases with different reactivities (White and Buss, 2014;Critelli et al., 2014;Kump et al., 2000). Beckingham et al. (2016) showed that the presence of a highly reactive volcanic glass phase indeed enhanced the chemical weathering rate of a volcanogenic sandstone. Deng et al. (2017) indicated that preferential dissolution of high reactivity phases enhances fracture evolution in carbonate rich shales. However, these experiments did not examine mechanical processes, such as grain detachment. Less reactive grains should become detached if their surrounding mineral matrix dissolves. Such complexity was in fact observed in bimineralic carbonate rocks, where relatively inert dolomite crystals detached from a calcite – dolomite comprised rock when their surrounding calcite matrix became dissolved (Levenson et al., 2017;Krklec et al., 2013). It remains unclear how the overall weathering rates, which include mechanical processes, is influenced by the proportion of high and low reactivity mineral phases.

In this study, we investigate how grain size and rock composition affect the chemical and mechanical weathering rate of rocks. To achieve this, we constructed a numerical model to simulate the weathering of rocks with a range of grain sizes, and different proportions of high reactivity and low reactivity minerals. Our model incorporates both chemical dissolution and grain detachment. In addition to exploring the impact of rock texture and reactivity on weathering rates, we evaluate the implications of the study to subsurface rock-fluid interactions.

## 2 Methods

### 2.1 Modeling rock weathering

We developed a numerical model in MATLAB to simulate the combined effects of chemical and mechanical weathering. A 2D cross section of a polycrystalline rock was represented by a domain (*560* elements wide and *420* elements in height) constructed with Voronoï tessellation (Taleb and Stafiej, 2011;di Caprio et al., 2016). Every Voronoï cell represents a rock grain comprising a single mineral phase, and the cell edges represent the inter-granular boundaries (Figure 1a). The range of grain size was varied in our simulations by dividing the domain into a different number of grains (100, 200, 400, 800,



1600, 3200, 6400) while keeping the number of elements constant. The grain sizes here are arbitrarily normalized to the size of the grains in the simulation with the largest number of grains (i.e., 6400), so that the relative grain size for the simulation with 6400 grains is 1, while the grain size for the simulation with 100 grains is 64. The impact of grain size on weathering was examined by comparing simulations of a coarse-grained rock with that of a rock comprising small grains (relative grain size equals 64 and 2 respectively).

equals 64 and 2 respectively).

To account for the heterogeneity of surface reactivity (Fischer et al., 2014), every element in the domain is assigned a characteristic value ($M_i$; Figure 1b) that corresponds to a fluid state ($M_F$), a high reactivity grain ($M_R$), a low reactivity grain ($M_L$), or a grain boundary ($B_{ii}$). There are three possible kinds of grain boundaries: boundaries between two high reactivity grains ($B_{RR}$), boundaries between two low reactivity grains ($B_{LL}$), and boundaries between one high reactivity and low

reactivity grain ($B_{RL}$). These values are reflective of the kinetic rate of dissolution, so that 1 represents the maximum rate of dissolution and 0 represents an inert state. In our formulation, grain boundaries dissolve one order of magnitude more rapidly than the mineral core, which is consistent with experimental work (Emmanuel, 2014;Bray et al., 2015;Lüttge et al., 2013).

To determine the impact of rock composition on weathering rate, we carried out simulations in which the proportion of low reactivity minerals within the rock is varied from 0% to 100%. Over 600 simulations of rocks with different

compositions were carried out.

In the initial state in our simulations, no elements are designated as fluid, although the rock is exposed to a reactive solution at the top boundary of the domain. We chose to simulate a system that is far from chemical equilibrium, which is representative of weathering rocks exposed at Earth's surface. To model the time dependent evolution of the rock, we used a cellular automaton approach, similar to that used by Taleb and Stafiej (2011) to model metal corrosion. Each element in our

domain was subject to well defined rules, although we used a stochastic approach to simulate the random nature of dissolution (Fischer et al., 2014). At each time step, the only elements that dissolved were those in direct contact with the solution. The probability for any given element to undergo dissolution was calculated from the product of 2 parameters: the characteristic value for each element ($M_i$) and the probability that the element would dissolve in that time step ($P_i$). This probability increased linearly with the number of fluid elements by which it was surrounded, so that a solid element in contact with a single fluid

element was assigned a $P$ value of 12.5%, determined from a random Gaussian distribution, while an element surrounded by 8 fluid elements was given a $P$ value of 100%. The number of surrounding elements was defined using the Moore neighbourhood method (range=1) (Lishchuk et al., 2011). Any element that underwent dissolution became a fluid and was assigned a characteristic value of $M_F$. At the end of each time step, regions that were completely surrounded by fluid detached from the surface and were replaced by a characteristic value of $M_F$.

To minimize boundary affects, a bounding box was defined in the center of the domain. In all the simulations, the distance of the bounding box from the sides and the base of the entire domain corresponded to the maximal grain diameter in the simulation with the largest grains. The simulations were terminated when all the elements within the calculation bounding box were assigned a fluid state ($M_F$ sites).



For each simulation, the main parameters that were recorded were: (i) the number of time steps until complete dissolution; (ii) the ratio of mechanical detachment to the overall weathering; and (iii) the total number of grain detachment events. The dimensionless weathering rate, $R^*$ was calculated by dividing the number of elements inside the bounding box by the total number of time steps.

## 3 Results and discussion

### 3.1 Impact of grain size on weathering rates

During the time dependent simulations, grain size was found to strongly impact the weathering rate of rocks. In a comparison between the weathering rates in fine-grained rocks and coarse-grained rocks, we found that fine-grained rocks weathered more rapidly (Figure 2a). Increasing the relative grain size from 2 to 64 slowed the weathering rate by 37%. This was due mainly to the higher rate of chemical weathering in the small grained rocks (Figure 2b), although in both cases the mechanical weathering component (Figure 2c) contributed at least 33% to the overall weathering rate. Even though the rate of mechanical weathering was similar for both grain sizes, we observed a higher frequency of detachments in fine-grained rocks, with tiny grains being detached during most time steps. By contrast, in the coarse-grained rock, mechanical detachments were less frequent, although much larger grains were removed during each event as is evident from the step-like function in Figure 2c, and from simulation snapshots (Figure 2d-e). The snapshot from the fine-grained rock simulation (Figure 2d) shows a higher density of grain boundaries (39%) than the coarse-grained rock (7%). This high density of reactive grain boundaries is the factor responsible for the increased rate of dissolution.

Although we found that fine-grained rocks weathered faster than coarse-grained rocks, the relationship between grain size and weathering rate was non-linear and appeared to show an exponential decline with increasing grain size. This relationship was observed in all the rock compositions that we examined (Figure 3a). In our simulations, above a certain size, the decrease in weathering rate slows significantly, suggesting that in real rocks the influence of grain size may be limited to relatively small scales. The transition we observed occurs when the density of reactive boundaries is less than ~20% of the entire domain. This seemingly high density of grain boundaries could be expected to occur in rocks with sub-micrometer grains and may therefore be relatively rare in natural systems.

Even though the chemical weathering rate is slower for rocks with large grains, the contribution of grain detachment to the overall weathering rate increases with grain size (Figure 3b). In coarse-grained rocks, detachment can contribute as much as 45% to the total weathering. This trend in mechanical weathering with increasing grain size (Figure 3b) shows an opposite trend to that of the overall rate (Figure 3a). The main reason for this is that in large grained rocks, the grains undergo less dissolution before detachment because of the lower density of grain boundaries.

There are both experimental and field based evidence to support the trends we observed in our simulations. Experimental observations by Kirstein et al. (2016) showed a preferential dissolution of small grains in micrite compared to





larger grained rocks. This was explained by the micritic structure that promotes the exposure of new, highly reactive grain boundaries. In addition, using vertical scanning interferometry (Lüttge et al., 2013;Fischer et al., 2012), micrite was found to dissolve an order of magnitude faster than single calcite crystals. Field based measurements presented by Emmanuel and Levenson (2014) also showed that micritic limestone weathered significantly faster than coarse-grainstone.

However, not all studies show that grain size affects the weathering rate significantly. In experiments on rocks with different textures (ranging from micrite to marble), Levenson et al. (2015) found no significant difference in the rates of calcite dissolution. In addition, simulations of electrochemical particle detachments by Taleb and Stafiej (2011) showed that the effect of grain size is insignificant at long reaction times. It may be that in these experiments and simulations, the grain sizes were in the region of the size independent weathering regime we identified in Figure 3a. To determine if this is indeed the case, future
experiments and field-based work on a wider range of grain sizes are necessary.

### 3.2 Impact of rock composition on weathering rate

       Our simulations show a non-linear relationship between rock composition and weathering rate (Figure 4, Figure 5a). We found that the presence of a small amount of a low reactivity phase in a high reactivity rock produced a disproportionally large reduction in the overall rate. This reduction in rate is due primarily to a decline in chemical weathering, although
mechanical weathering also slows over the same range of compositions (Figure 4). However, when the low reactivity phase becomes dominant, the further addition of low reactivity grains has no significant impact on both the chemical and mechanical rate. We suggest that this trend is a result of the increasing density of low reactivity grain boundaries: as more low reactivity grains are added; the slow dissolution of their boundaries retards the rate at which new reactive grain boundaries are exposed to the fluid.

In contrast to the mechanical rate, which decreases as the amount of low reactivity phase increases (Figure 5a), the relative contribution of mechanical grain detachment to the overall weathering shows a more complex trend (Figure 5b); as the proportion of low reactivity grains increases from zero to 40% the contribution of grain detachment increases from 37% to 46%. When low reactivity grains make up 40-60% of the rock, the mechanical contribution reaches a maximum with values as high as 50%. However, as the proportion of low reactivity grains increases beyond 60% the mechanical contribution drops
linearly to a value of about 41%.

       The peak in mechanical weathering between 40-60% is not a result of an increasing number of detachment events in this range. This can clearly be seen at Figure 5c that shows that in fact, a minimum number of events occurs in this range. Instead, the observed trend can be explained by the size of detached clusters. The number of detachment events is greater than 2000 while the number of grains in the simulation is only 100, demonstrating that most of the detachments are sub-grains. In
the snapshots of rocks with different compositions it can be seen that when the rock comprises mainly high reactivity minerals, numerous detachment events occur, and the removed clusters are relatively small (Figure 6a). When the proportion of low reactivity minerals increases to 40-60% (Figure 6b) the rapid dissolution of the high reactivity phase leaves behind a skeleton of large clusters of low reactivity grains. It is the removal of these large clusters that increases the contribution of mechanical





weathering. The detachment of small clusters and individual grains again becomes the dominant mode when the rock comprises mainly low reactivity minerals (Figure 6c).

Although previous studies have shown that chemical dissolution rate is determined by the abundance of lower reactivity phases in a rock (White and Brantley, 2003;Wigley et al., 2013;Critelli et al., 2014), we show that the combined effects of chemical and small-scale physical processes control the overall weathering rate. Importantly, there is evidence from both field and laboratory experiments to support this. For example, Deng et al. (2017) showed that the presence of a high reactivity phase (such as calcite) strongly affected the stability of fractured shale surfaces, reporting that erosion was initiated only when the high reactivity phase exceeded 35%. Krklec et al. (2016) invoked a combination of chemical and micron scale physical proccesses to explain the results of weathering experiments on carbonate tablets. They suggested that most of the material loss was due to grain detachment that was triggered by dissolution along crystal edges. In a laboratory study of fine grained limestone, Levenson and Emmanuel (2016) reported that grain detachment accounted for approximately 38% of the material removed from the rock surface, which is within the range (37-50%) predicted by our model. However, the precise proportion of mechanical weathering for any given rock is likely to be strongly dependent on the rate of dissolution along grain edges.

### 3.3 Effect of the rate of dissolution along grain boundaries on weathering

The last set of simulations was aimed at examining how the rate of dissolution along grain boundaries impacts the weathering process. In addition to the existing model in which the grain boundaries reacted at a rate that was dependent on its mineral phase (which we define here as the heterogeneous model), we constructed an alternative model in which all the grain boundaries reacted at the same rate (defined as the homogeneous model). A comparison of the two kinds of simulations shows that weathering rate of rocks with homogeneous grain boundaries is almost constant, regardless of the mineral composition (Figure 7a). This is due to higher density of reactive grain boundaries (relative to the heterogeneous model) that enhances the contribution of mechanical weathering to the overall rate (Figure 7b). The proportion of grain detachment in the homogeneous model shows a linear increase with increasing proportion of low reactivity minerals, reaching a maximum value of 86%. This result shows that the reactivity of the grain boundaries is the principal factor in determining whether or not mechanical detachment is the dominant mode of weathering. This is a result of the rapid dissolution of highly reactive grain boundaries which promotes grain detachment, and effectively compensates for the increasing presence of low reactivity minerals. In addition, the homogeneous model also shows a dramatic decrease in the number of detachment events with increasing proportion of the low reactivity phase (Figure 7c). This reflects the transition from the detachment of tiny sub-grains to the detachment of large complete grains.

### 4 Conclusions

In this study, we used 2D cellular automaton simulations to examine the impact of grain size and rock composition on chemical weathering and chemo-mechanical grain detachment. While our results indicate that the weathering rate decreases





with increasing grain size, we also found that the weathering rate shows a nonlinear dependence on the proportion of low reactivity minerals. This is due to the influence of chemical weathering which is mainly controlled by the density of grain boundaries. In addition, we found that the contribution of mechanical weathering to the overall weathering is determined by the rate of dissolution along grain boundaries.

Our model is the first step towards accurately simulating the coupling between physical and chemical processes during weathering in rocks. As such, there are clearly a number of limitations. Our simulations do not include standard kinetic formulations, or physical processes, such as diffusion and fluid flow. In addition, little is known about the rates of dissolution along grain boundaries for different mineral systems, so that it cannot yet be determined whether the heterogeneous model or the homogeneous model is more realistic. Future work focusing on the development of more sophisticated models, and the
experimental measurement of dissolution rates along grain boundaries, may provide a way to resolve these issues.

       In addition to providing insight into the feedback between mechanical and chemical processes during weathering, our results also have implications for fluid-rock interaction in the subsurface. Enhanced oil recovery, hydraulic fracturing, and $CO_2$ sequestration all involve the injection of reactive fluids into rock formations. The selective dissolution of high reactivity minerals during such operations could result in the detachment of mineral particles, potentially impacting rock porosity and
permeability. Furthermore, if these mineral phases contain toxic elements or compounds, grain detachment could represent an important mode of mobilizing contaminants in groundwater. An example of such a system is carbonate-rich shales. Such shales often contain pyrite which is a potential source of arsenic contamination. The pyrite in these rocks, however, is much less reactive than the calcite. Exploring the way pyrite particles are mobilized in shale as a result of calcite dissolution is an ongoing avenue of our research.

**5 Acknowledgments**

       This research was supported by a student scholarship from the Ministry of National Infrastructures, Energy, and Water Resources and by The Hebrew University Center for Nanoscience and Nanotechnology. The Israel Science Foundation is thanked for their support.

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



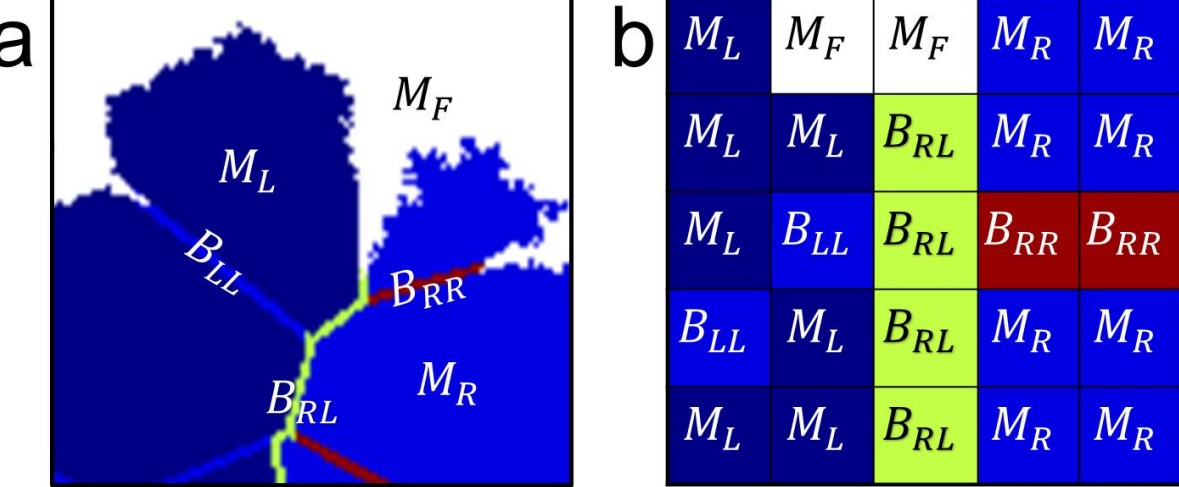

Figure 1: Part of the domain simulating a rock in the model at (a) the macro scale and (b) the pixel scale. The rock is simulated as a 2D cross section using Voronoï tessellation. Grains are represented as Voronoï cells containing a mineral phase and cell edges represent the grain boundaries. Every element is given a characteristic value of either a fluid state ($M_F$, white), a high reactivity grain ($M_R$, blue), a low reactivity grain ($M_L$, dark blue), a grain boundary between two high reactivity grains ($B_{RR}$, red), a boundary between two low reactivity grains ($B_{LL}$, blue), or a boundary between one high reactivity and one low reactivity grain ($B_{RL}$, green).





**Figure 2: Number of removed elements as a function of time steps for 2 different simulations. Elements removed by (a) total weathering; (b) chemical weathering; and (c) mechanical weathering. The black lines represent the simulation with small grains (relative grain size=2), while the blue lines represent the number of removed elements in the simulation with the large grains (relative grain size =64). The camera icon in (a) indicates the timing of the snapshots for the (d) small grain simulation and (e) the large grain simulation. The domain is comprised entirely of high reactivity grains. For both grain sizes, chemical weathering contributes more to the total weathering than mechanical weathering. In the fine-grained rock (d), 67% of the material has been removed; by contrast, in the snapshot from the coarse-grained rock (e) taken after the same number of time steps, only 42% of the rock has been removed. The coarse-grained rock has a 7% density of grain boundaries, while in the fine-grained rock the boundaries cover 39% of the simulated rock area. The higher density of reactive grain boundaries seen in the fine-grained rock promotes the increased rate of dissolution.**



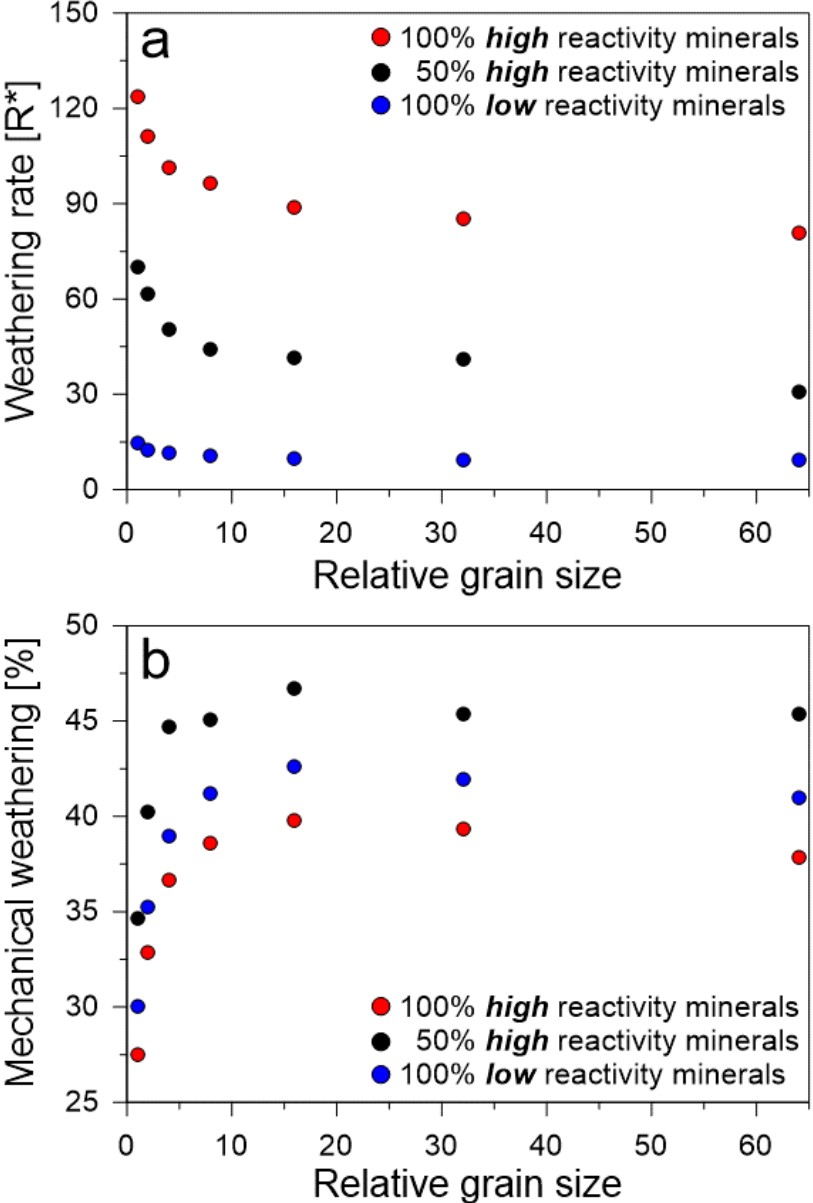

**Figure 3: Effect of grain size on weathering for three different compositions: (a) weathering rate versus relative grain size in units of R\*, and (b) the ratio of mechanical weathering to overall weathering versus relative grain size. Red circles represent simulations for a rock comprising 100% high reactivity mineral phase. Black circles represent an equal mix of high and low reactivity mineral phases. Blue circles represent a rock comprising 100% low reactivity mineral phases. Weathering rate decreases strongly with relative grain size until a plateau is reached at larger grain sizes. Similarly, the mechanical contribution also increases with increasing grain size until a plateau is reached.**



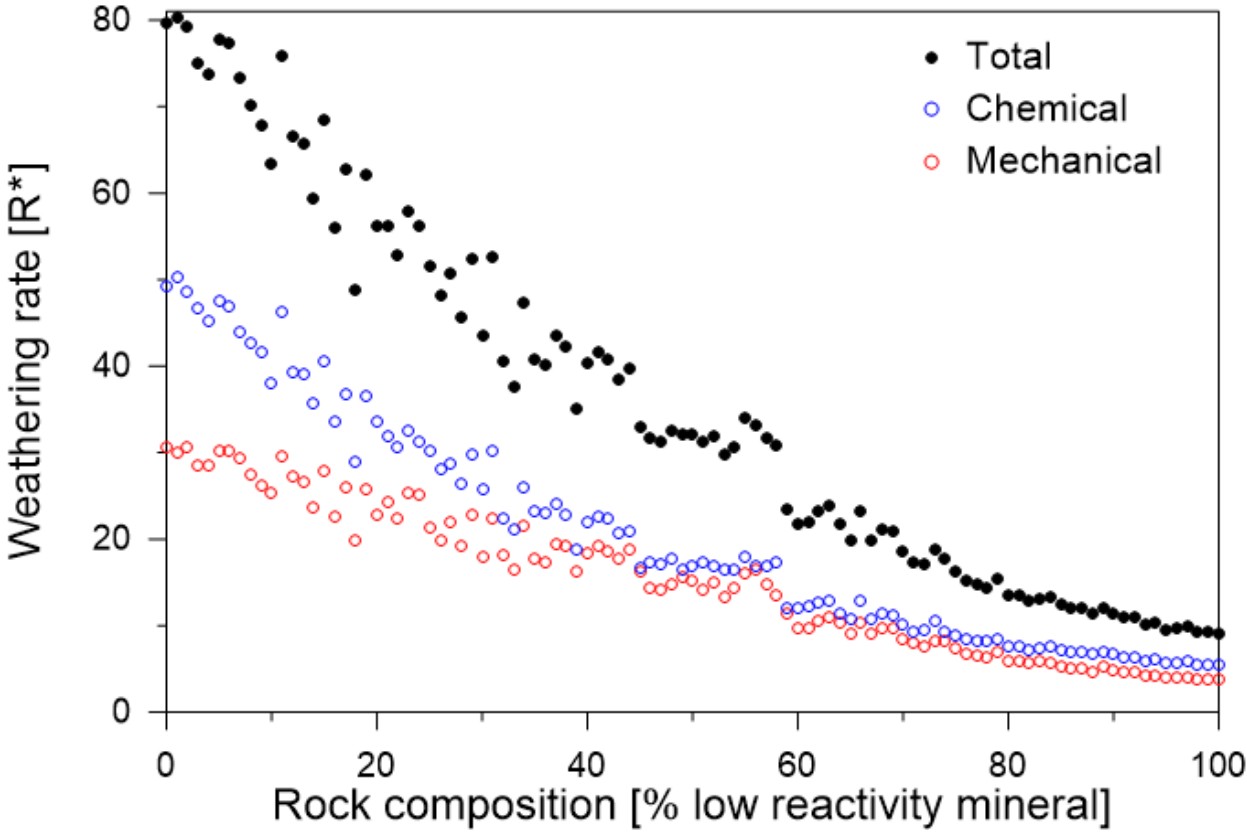

**Figure 4: Impact of rock composition on weathering rate.** The overall weathering rate (full black circles) decreases with the addition of low reactivity minerals to the rock. Chemical weathering (hollow blue circles) shows a disproportionally large decrease with a small addition of low reactive phases. Mechanical weathering (hollow red circles) shows a more uniform decrease with the change in the rock composition, indicating that the main factor controlling the overall weathering rate is the chemical weathering rate.




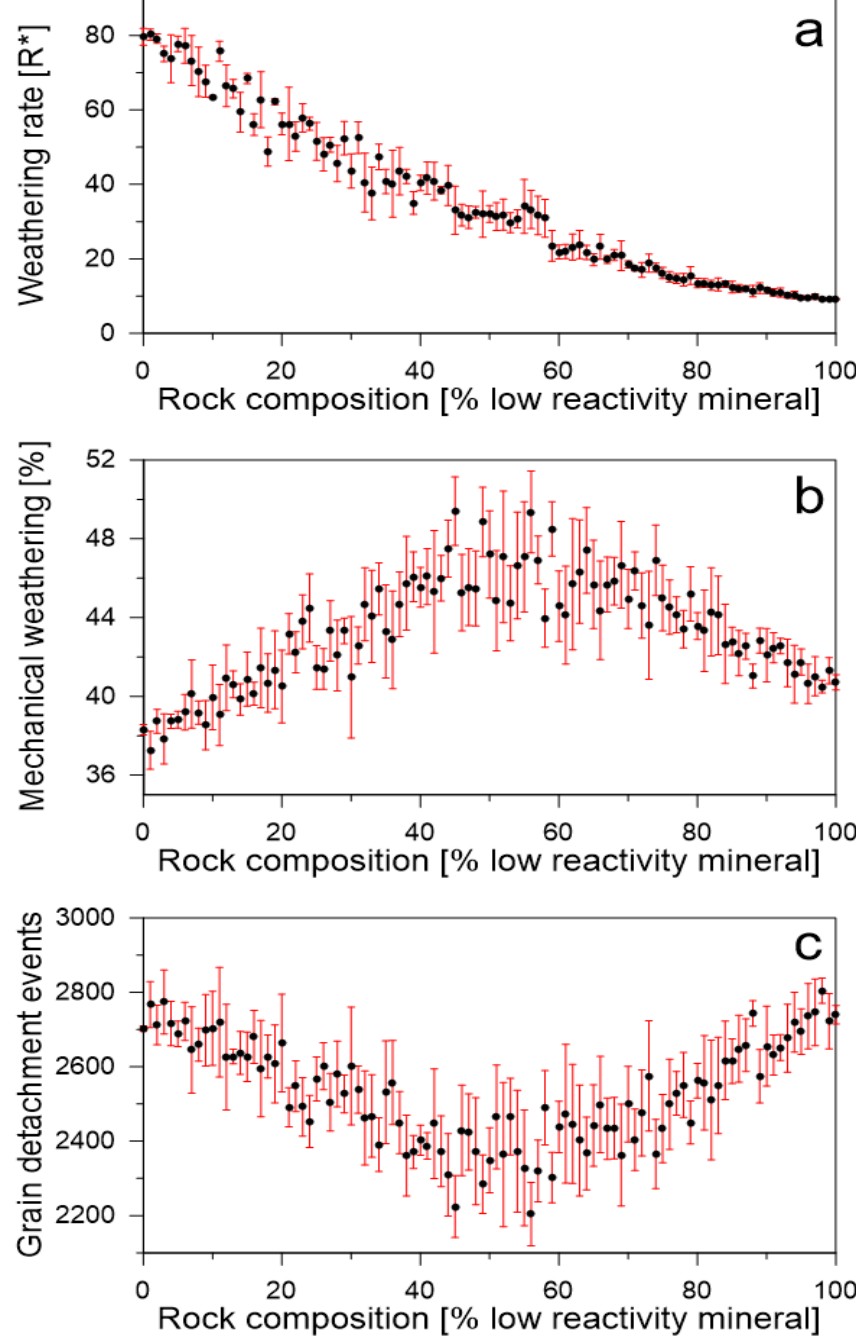

**Figure 5: Impact of rock composition on weathering. (a) Weathering rate in R\* units versus composition; (b) ratio of mechanical weathering to total weathering as a function of composition; and (c) the number of detachment events versus composition. Black circles show the average of 6 different realizations for each composition, while the red bars show the standard deviation. The stochastic nature of dissolution and the random position of low reactivity grains in the simulated rocks, causes variation in the weathering rate between rocks with the same composition. The weathering rate decreases non-linearly with the addition of low reactivity minerals. When there are approximately equal proportions of high and low reactivity minerals, mechanical weathering reaches a maximum, although the actual number of detachment events is at a minimum.**





**Figure 6: Simulation snapshots for three representative compositions. Rocks comprising (a) 10% low reactivity minerals; (b) an equal mix of high and low reactivity minerals; and (c) 90% low reactivity minerals. In all simulations, the grain size is constant (relative grain size = 64). Orange arrows indicate clusters that are removed via mechanical grain detachment. Small sized grains are detached in (a) and (c); in contrast, large clusters detach in (b). Note that although (a) and (c) have similar granular structures, the overall weathering rate in (c) is almost an order of magnitude slower.**





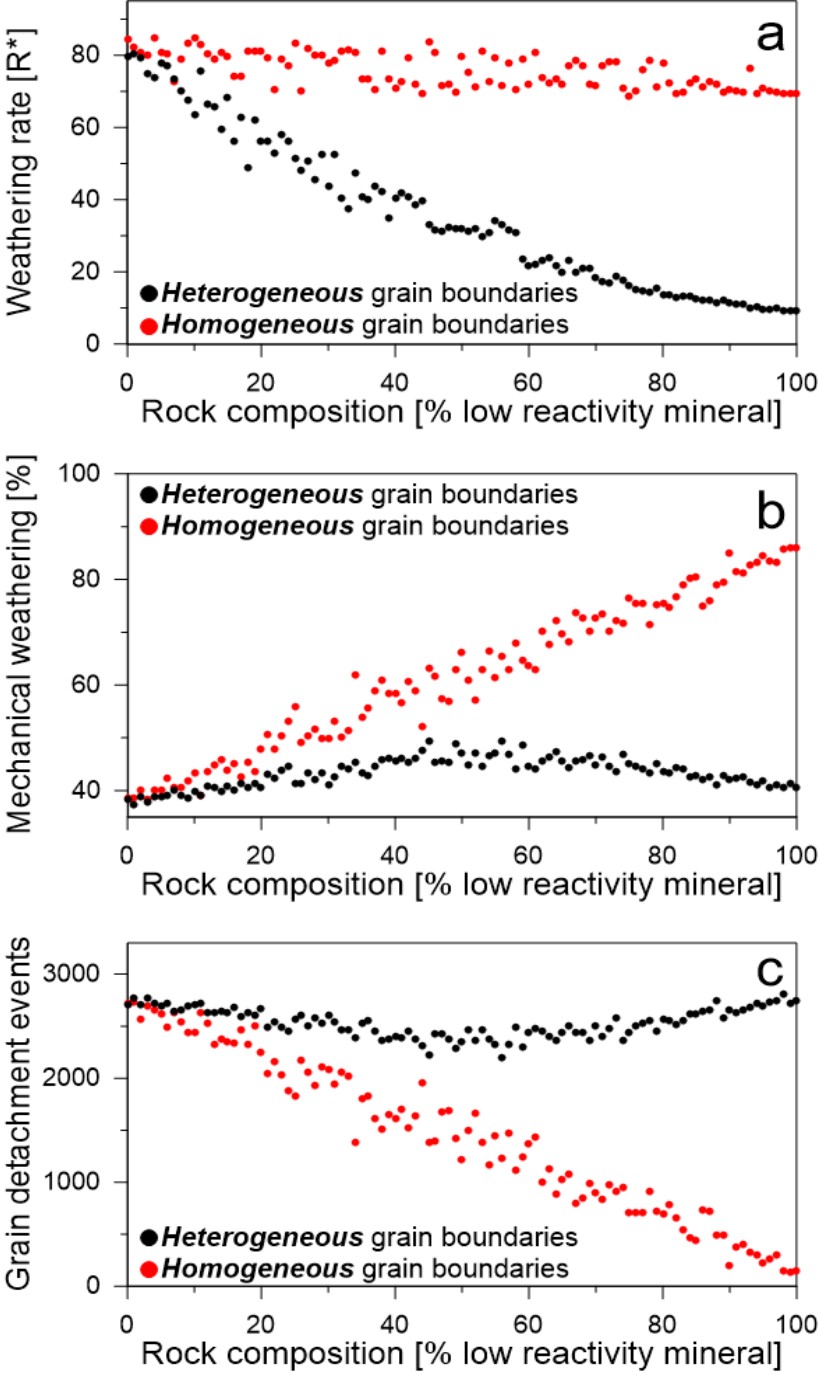

**Figure 7: Impact of rock composition on weathering in heterogeneous and homogeneous grain boundary models. (a) Weathering rate in R\* units versus composition; (b) ratio of mechanical weathering to total weathering as a function of composition; and (c) the number of detachment events versus composition. The heterogeneous model assumes that the reactivity for each grain boundary is dependent on the type of the adjacent minerals, while in the homogeneous model all boundaries have the same reactivity.**