# Peer review of "Impact of grain size and rock composition on simulated rock weathering"

_Earth Surface Dynamics, 2018_

## Referee Comment (RC1) · K. Krklec (Referee) · 22 Jan 2018

**Impact of grain size and rock composition on simulated rock weathering**

**Yoni Israeli and Simon Emmanuel**

**General comments:**

This manuscript presents the use of a numerical model to assess the effect of grain size on chemical weathering and chemo-mechanical grain detachment. Simulation of rock weathering takes into account a range of grain sizes, and different proportions of high reactivity and low reactivity minerals. Results indicate that the weathering rate decreases with increasing grain size, as well as nonlinear dependence on the proportion of low reactivity minerals. This is a very interesting research with interesting conclusions and hopefully future studies will confirm (or reject) these findings.

**Specific comments:**

In Methods section authors describe dissolution (chemical weathering), but it is not clear how mechanical weathering is calculated. Authors do give "hint" of that explanation in Page 3, Line 28-29 and in the last paragraph of Methods section (Page 4, Lines 1-5), but this should be described more clearly.

In the last paragraph of Conclusions section (Page 7, Lines 11-19) authors mention implications of their research. Although it is encouraged to mention this in Conclusions with intention to describe/direct future research it is too extensive and general. Rephrased version of this paragraph should be included in Introduction section, accompanied with references.

**Technical comments:**

References in text should be listed chronologically. As well, in References section abbreviated journal names should be used.

Line 20: Wilson, 2004 – reference is not listed in the References. I presume this is the reference for weathering of architectural stone. If not, please provide additional reference.

Line 13: Correct CO2 to $CO_2$

Thank you for the opportunity to review!

Kristina Krklec

---

## Short Comment (SC1) · 22 Jan 2018

I'm excited by the type of work described in this paper and it's a topic that needs much more attention in our field. The model that the authors present predicts that "as grain size increases, the weathering rate initially decreases" seems to contradict some of my observations in the Sierra Nevada (CA). The Sierras are a patchwork of different plutonic rocks and, oftentimes, the differential erosion between the fine-grained rocks and adjacent coarse-grained rocks is striking, with the latter seeming to resist weathering and erosion much better than the former. To be fair, these observations have not controlled for mineralogy. Have the authors found field examples that support their model's results? If not, what aspect of weathering might the model be missing?

---

## Referee Comment (RC2) · C. Fischer (Referee) · 2 Mar 2018

Review of the article "Impact of grain size and rock composition on simulated rock weathering" by Israeli and Emmanuel Cornelius Fischer Helmholtz-Zentrum Dresden-Rossendorf, Institut f. Ressourcenökologie, Abteilung Reaktiver Transport (Leipzig)

The paper by Israeli and Emmanuel focuses on an interesting and important topic, i.e., the impact of grain size and crystal reactivity on rock weathering. The authors present an interesting pool of data based on simulation results. They include into the simulation heterogeneous material that mimics two types of domains and of domain boundaries that are characterized by high vs. low reactivity, respectively. In doing so, the simulation results are analyzed with respect to the material loss over time, proportional to

the reaction rate. The combined parameter variation of grain size and surface reactivity is then utilized to get deeper insight into the importance of chemical weathering (dissolution) vs. chemo-mechanical weathering (dissolution plus grain detachment). As the authors already state in the manuscript, the reacting grains or domains do not show any internal heterogeneity such as defect structures etc. The contrast in defect density would add additional constraints to the evolution of material flux from the reacting surface [1]. Nevertheless, the presented results add important quantitative insight that can be utilized for several applications, or for the explanation of quantitative results of case studies. The specific case of different types of minerals is highlighted in the paper already. Moreover, this study continues the detailed previous observations of the authors about the phenomenon of chemo-mechanical rock weathering [2]. The main results of the present study towards this question are new interpretations about geometric constraints quantifying the impact of grain size on the overall rock weathering rate due to chemo-mechanical weathering. The authors discuss in the paper how such impact is controlled by so-called reactive boundaries and their density in the material under investigation. Additionally to the thought about contrasting defect density mentioned above, one could additionally think about interpretations that highlight contrasting and preferred crystal orientation, implemented by the presented approach of several domains. In that sense, the geometric approach highlighted in this manuscript offers potential for multiple applications in the weathering community.

References: [1] Fischer, C., Kurganskaya, I., Luttge, A., 2018. Inherited control of crystal surface reactivity. Applied Geochemistry 91, 140-148. [2] Emmanuel, S., Levenson, Y., 2014. Limestone Weathering Rates Accelerated by Micron-Scale Grain Detachment. Geology 42, 751-754.

**ESurfD**

---

## Author Comment (AC1) · 11 Mar 2018

Yoni Israeli and Simon Emmanuel

swemmanuel@gmail.com

SC 1.1: The model that the authors present predicts that "as grain size increases, the weathering rate initially decreases" seems to contradict some of my observations in the Sierra Nevada (CA). The Sierras are a patchwork of different plutonic rocks and, oftentimes, the differential erosion between the fine-grained rocks and adjacent coarse-grained rocks is striking, with the latter seeming to resist weathering and erosion much better than the former. To be fair, these observations have not controlled for mineralogy.

AC 1.1: Thank you for your feedback. As you pointed out the coarse-grained plutonic rocks at the Sierra Nevada resist weathering and erosion much better than the fine-grained rocks. This is in fact what our simulations predict for part of the size range

that we explored (In the revised manuscript page 4 lines 22-32). In addition, our model shows that the impact of mineralogy on weathering rate is much larger than the impact of grain size (see figure 3a). It might be interesting to carry out a more quantitative comparison of the rocks mentioned in the Sierra Nevada.

SC 1.2: Have the authors found field examples that support their model's results? If not, what aspect of weathering might the model be missing?

AR 1.2: We have found some evidence for the dependence of weathering rate on grain size and this is discussed in the revised manuscript on page 5 lines 13-24. We entirely agree that future experiments and field-based work on a wider range of grain sizes are necessary (page 7 lines 26-27). We also discuss the limitations of our model on page 7 lines 20-25.

---

## Author Response (AR1)

**Response to Kristina Krklek referee comments on "Impact of grain size and rock composition on simulated rock weathering" by Yoni Israeli and Simon Emmanuel**

We thank Kristina Krklek for her helpful and constructive review. Responses to the comments are provided below, in underlined italics.

**RC 1.1:** In Methods section authors describe dissolution (chemical weathering), but it is not clear how mechanical weathering is calculated. Authors do give "hint" of that explanation in Page 3, Line 28-29 and in the last paragraph of Methods section (Page 4, Lines 1-5), but this should be described more clearly.
**AC 1.1:** *After each time step the rock is scanned for regions that reached the threshold for chemo-mechanical detachment, i.e. grains that are surrounded by fluid from all directions. These regions are replaced to a characteristic value of $M_F$. The number of elements that underwent mechanical grain detachment and the number of detachment events is updated after each time step. In the revised manuscript we clarify this point on page 3, lines 7-10.*

**RC 1.2:** In the last paragraph of Conclusions section (Page 7, Lines 11-19) authors mention implications of their research. Although it is encouraged to mention this in Conclusions with intention to describe/direct future research it is too extensive and general. Rephrased version of this paragraph should be included in Introduction section, accompanied with references.
**AC 1.2:** *Since the original manuscript was submitted, the specific example we gave of the research on pyrite detachment in shales was published in EPSL (Kreisserman and Emmanuel, 2018). We believe that the model could be used to simulate the mobilization of contaminants in a way that cannot be achieved using existing geochemical models. In the revised manuscript this point is discussed in more detail (page 2, lines 22-34, page 7, lines 28-32, page 8, lines 1-2), and some additional references have been included.*

**RC 1.3:** References in text should be listed chronologically. As well, in References section abbreviated journal names should be used.
**AC 1.3:** *In the revised manuscript we have made sure that all the references in the text are in chronological order and have been cited according to the journal format.*

**RC 1.4:** Page 1, Line 20: Wilson, 2004 – reference is not listed in the References. I presume this is the reference for weathering of architectural stone. If not, please provide additional reference.
**AC 1.4:** *In the revised manuscript we have added the citation for Wilson, 2004 to the reference list. We have included an additional reference (page 9, line 37) concerning the weathering of architectural stone (Price, C. A., & Doehne, E. ,2011).*

**RC 1.5:** Page 7, Line 13: Correct CO2 to CO2
**AC 1.5:** *This has been corrected (page 2, line 23).*

**Response to Cornelius Fischer referee comments on "Impact of grain size and rock composition on simulated rock weathering" by Yoni Israeli and Simon Emmanuel**

We thank Cornelius Fischer for his helpful and constructive review. Responses to the comments are provided below, in underlined italics.

**RC 2.1**: As the authors already state in the manuscript, the reacting grains or domains do not show any internal heterogeneity such as defect structures etc. The contrast in defect density would add additional constraints to the evolution of material flux from the reacting surface [1].

**AR 2.1:** *We agree that both intragranular and intergranular heterogeneity have the potential to affect both reaction rates and dissolution patterns. In the revised manuscript this is discussed on page 7 lines 24-27 and represents a promising direction for future research.*

**RC 2.2:** Additionally to the thought about contrasting defect density mentioned above, one could additionally think about interpretations that highlight contrasting and preferred crystal orientation, implemented by the presented approach of several domains. In that sense, the geometric approach highlighted in this manuscript offers potential for multiple applications in the weathering community.

**AR 2.2:** *We thank the reviewer for pointing out this implication, and we discuss it in the revised manuscript page 7, lines 28-32, page 8 lines 1-2.*

**Impact of grain size and rock composition on simulated rock weathering**

Yoni Israeli and Simon Emmanuel

[revised manuscript text omitted]
.  Moreover, crystals often contain defects so that their reactivity may in fact be heterogeneous, which could affect both reaction rates and dissolution patterns (Fischer et al., 2018). Future work focusing on the development of more sophisticated models, and the experimental measurement of dissolution rates along grain boundaries, may provide a way to resolve these issues.

 In addition to shedding light on the behaviour of rocks comprising different minerals, our modelling approach could also be used to simulate rocks with different crystallographic orientations. Different surfaces of the same crystal often react at different rates (Godinho et al., 2014), so that the two different mineral reactivities in our model could represent a rock containing one mineral phase with two different orientations. In that sense, an advantage of using such a model is that it can be modified to represent various kinds of rocks and provide insight into effects of both mineral orientation and spatial distribution on mechanical and chemical processes during their weathering. In addition, the model could provide a way to simulate the mobilization of particles and contaminants in a way that cannot be achieved using existing geochemical models.

~~The selective dissolution of high reactivity minerals during such operations could result in the detachment of mineral particles, potentially impacting rock porosity and permeability. Furthermore, if these mineral phases contain toxic elements or compounds, grain detachment could represent an important mode of mobilizing contaminants in groundwater. An example of such a system is carbonate-rich shales. Such shales often contain pyrite which is a potential source of arsenic contamination. The pyrite in these rocks, however, is much less reactive than the calcite. Exploring the way pyrite particles are mobilized in shale as a result of calcite dissolution is an ongoing avenue of our research.~~

**5 Acknowledgments**

This research was supported by  student scholarships from the Ministry of National Infrastructures, Energy, and Water Resources and  The Hebrew University Center for Nanoscience and Nanotechnology. The Israel Science

Foundation is thanked for additional financial support. We also thank Kristina Krklec and Cornelius Fischer for their constructive reviews.

[revised manuscript text omitted]